# Improving Site-Specific Maize Yield Estimation by Integrating Satellite Multispectral Data into a Crop Model

**Vijaya R. Joshi [1]**, **Kelly R. Thorp [2]**, **Jeffrey A. Coulter [1]**, **Gregg A. Johnson [1,3]**, **Paul M. Porter [1]**, **Jeffrey S. Strock [4,5]** and **Axel Garcia y Garcia [1,5,*]**

[1] Department of Agronomy and Plant Genetics, University of Minnesota, St. Paul, MN 55108, USA; joshi232@umn.edu (V.R.J.); jeffcoulter@umn.edu (J.A.C.); johns510@umn.edu (G.A.J.); pporter@umn.edu (P.M.P.)

[2] U.S. Arid Land Agricultural Research Center, United States Department of Agriculture-Agricultural Research Service, Maricopa, AZ 85138, USA; kelly.thorp@ars.usda.gov

[3] Southern Research and Outreach Center, University of Minnesota, Waseca, MN 56093, USA

[4] Department of Soil, Water, and Climate, University of Minnesota, St. Paul, MN 55108, USA; jstrock@umn.edu

[5] Southwest Research and Outreach Center, University of Minnesota, Lamberton, MN 56152, USA

[*] Correspondence: axel@umn.edu; Tel.: +1-507-752-5080; Fax: +1-507-752-5097

**Abstract:** Integrating remote sensing data into crop models offers opportunities for improved crop yield estimation. To compare site-specific yield estimation accuracy of a stand-alone crop model with a data-integration approach, a study was conducted in 2016–2017 with nitrogen (N)-fertilized and unfertilized treatments across a heterogeneous 7-ha maize field. For each treatment, yield data were grouped into five classes resulting in 109 spatial zones. In each zone, the Crop Environment Resource Synthesis (CERES)-Maize model was run using the GeoSim plugin within Quantum GIS. In the data integration approach, maize biomass values estimated using satellite imagery at the five (V5) and ten (V10) leaf collar stages were used to optimize the total soil nitrogen concentration (SLNI) and soil fertility factor (SLPF) in CERES-Maize. Without integration, maize yield was simulated with root mean square error (RMSE) of 1264 kg ha$^{-1}$. Optimization of SLNI improved yield simulations at both V5 and V10. However, better simulations were obtained from optimization at V10 (RMSE 1026 kg ha$^{-1}$) as compared to V5 (RMSE 1158 kg ha$^{-1}$). Optimization of SLPF together with SLNI did not further improve the yield simulations. This study shows that integrating remote sensing data into a crop model can improve site-specific maize yield estimations as compared to the stand-alone crop modeling approach.

**Keywords:** spatial-variability; precision agriculture; site-specific calibration; crop modeling; remote sensing

## 1. Introduction

Early estimation of crop yield is important to farmers, government agencies, and policy makers in improving crop production efficiency and detecting potential biotic and abiotic risks that affect crop yield [1,2]. Mounting pressures to address environmental problems resulting from crop production [3,4] and increasing competition for greater economic efficiency [5] have directed research efforts for site-specific crop yield estimation. Spatially explicit estimation of crop yield not only helps to explain the spatial variability of crop growth within a field but also to optimize crop management efforts and reduce risks [6]. In recent decades, technological advancement in satellite-based global positioning systems (GPS), improved sensor capabilities, computational tools,

and geographic information systems (GIS) have greatly enabled digital data-driven approaches for site-specific crop yield estimation [7–10]. Frequently used approaches include yield maps [11,12], remote sensing images [13–15], and process-based crop simulation models [16,17].

Sensors mounted on combine harvesters calculate the mass of grain per unit of area harvested, which together with GPS receivers provide grain yield measurements at geo-referenced points to produce yield maps that are effective in visualizing spatial variability of crop yield [18]. Historical yield maps help to locate high and low yielding regions within a field and are useful in estimating site-specific yield [12,19]. However, yield maps alone are less revealing regarding the cause of yield variation [18]. Furthermore, historical yield maps can vary widely between years due to differences in growing season weather, making future in-season yield estimates challenging [20,21]. Remote sensing images, in this regard, are advantageous for monitoring in-season crop growth patterns in response to the effects of weather, pests, disease, and other management issues.

Remote sensing devices, such as multispectral and hyperspectral sensors, when mounted on aerial or satellite platforms can cover large areas and can give rapid assessment of within-field variability of crop growth. Spectral indices, such as the normalized difference vegetation index (NDVI) [22,23], obtained from remote sensing images correlate with crop growth status and help to estimate site-specific crop yield [13,14,19,24]. Additionally, time-series spectral data from remote sensing devices are also used to develop models to characterize temporal change in crop growth, such as leaf area index, which have been found to be promising for crop yield estimation [15]. Despite distinguishing crop growth and yield variability and being advantageous at both temporal and spatial scales, remote sensing images alone, similar to yield maps, have limitations in determining yield-limiting factors, such as nutrients, water, or pests.

Process-based crop models incorporate soil, weather, cultivar, and crop management information to simulate crop growth and yield [25,26]. Therefore, crop models have an advantage over yield maps and remote sensing images by helping to identify the sources of yield variation in a way that can be used to optimize crop management efforts. The major limitation of crop models comes from their point-based nature. Since crop models typically only incorporate input data from one point in space, they have reduced capability at larger spatial scales. For example, heterogeneity in soil properties across a field requires extensive parameterization for larger spatial application of crop models [17]. Due to differences in field topography and soil properties, events such as runoff are inevitable and can change soil moisture conditions, nitrogen (N) levels, and ultimately yield. The point-based nature of crop models does not account for such spatial processes of runoff from adjoining regions [16]. High resolution soil sampling, in terms of space and time, can detect spatial processes but can be impractical. Additional constraints for crop model applications are the requirement of multiple site-years of field-collected data on a particular cultivar for calibration and evaluation purposes. In practical situations, multiple sites and years of field data are not always available for detailed calibration and evaluation. In addition, new cultivars are released frequently, making any previous calibration efforts obsolete. In such data-scarce situations, one option is to integrate soil and crop parameters obtained from remote sensing images into crop models. As satellite or aerial images have the advantage of rapid, site-specific estimation of soil and crop bio-physical parameters across larger spatial scales, they offer great opportunities to address the limitations of crop models by supporting spatial modeling applications and facilitating spatial calibration of models [16,17,27].

Numerous attempts have been made at integrating remote sensing data into point-based crop models for site-specific crop yield estimation [17,28–30]. Primary schemes of data integration have been to replace, update, re-initialize, or optimize the model inputs that are missing or difficult to measure across the field [27,31–33]. For example, Dente et al. [29] integrated leaf area index (LAI) estimated from remote sensing images into the Crop Environment Resource Synthesis (CERES)-Wheat model to map wheat (*Triticum aestivum* L.) yield variability in Southern Italy at the catchment scale to optimize sowing date, wilting point, and field capacity parameters, which improved yield estimates as compared to a no-integration approach. Ines et al. [34] integrated moderate resolution imaging

spectroradiometer (MODIS) LAI and remotely estimated soil moisture in the CERES-Maize model, obtaining improved maize (*Zea mays* L.) yield simulations compared to stand-alone crop model outputs. More recently, Ban et al. (2019) [30] also integrated MODIS-derived seasonal LAI as well as water stress factors into the CERES-Maize model for improved accuracy of maize yield estimations. Similar improvements with the use of data integration have been reported by Guo et al. [35], Launay and Guerif [32], and Li et al. [36]. Although data integration has been shown to be an effective method for enhancing site specific yield estimation, most of the previous conclusions have been drawn from state- or regional- level studies [28–30,34] or plot-scale experiments [36]. Limited studies have been carried out to address within-field variability [35,37]. Studies at the field scale are important to examine the efficacy of data-integration for site-specific crop production optimization and for precision management of agricultural inputs. Besides, previous studies on data integration have frequently used LAI as the state variable to connect remote sensing and crop modeling [28,29,34,36,37]. Surprisingly, crop parameters other than LAI, such as biomass and canopy N content, have been less explored. Additionally, time series measurements of crop parameter have been used in past studies, with little attention given to early growth stage measurements. Options for crop management are restricted at later growth stages (late vegetative to reproductive stages). Studies on data integration at early stages of crop development will aid in decision making, for example in precision application of fertilizers and irrigation. Despite several studies, integration of remote sensing data into crop models is not yet a common practice and several details of the integration processes still need to be worked out. Some of the crucial information that is lacking includes the identification of a process to correlate remote sensing measurements to state variables of crop models, improvement of data integration methods, and implementation of the integration process to several aspects of crop production. The objectives of this study were to (i) develop an approach for improved maize yield estimation by integrating multispectral data of maize canopies at early vegetative stages into a crop growth model and (ii) compare site-specific yield estimation accuracy of a stand-alone crop model with a data integration approach, where soil parameters in the crop model were spatially optimized from satellite images derived biomass at five (V5) and ten leaf collar (V10) stages.

## 2. Materials and Methods

### 2.1. Study Site and Year

The field study was conducted during the growing seasons of 2016 and 2017 at the University of Minnesota Southwest Research and Outreach Center, located near Lamberton, MN, USA (44°14′19″ N and 95°18′50″ W) at an elevation of about 350 m ASL. The dominant soil type is Normania clay loam (Fine-loamy, mixed, superactive, mesic Aquic Hapludoll) [38]. The region has a hot summer humid continental climate.

### 2.2. Field Experiment and Data Collection

The experiment was carried out across a heterogeneous 7-ha field in strips with and without N fertilizer; the fertilized strips received sidedress applications of urea ammonium nitrate at the six leaf collar stage (V6) at the rate of 118 and 135 kg N ha$^{-1}$ in 2016 and 2017, respectively. Four replications of the two treatments were randomized in a complete block design. The maize hybrid "Pioneer P0297" was planted with a row spacing of 0.76 m for a target population of 88,900 plants ha$^{-1}$. Each strip contained a minimum of 18 rows and was at least 230 m long (Figure 1).

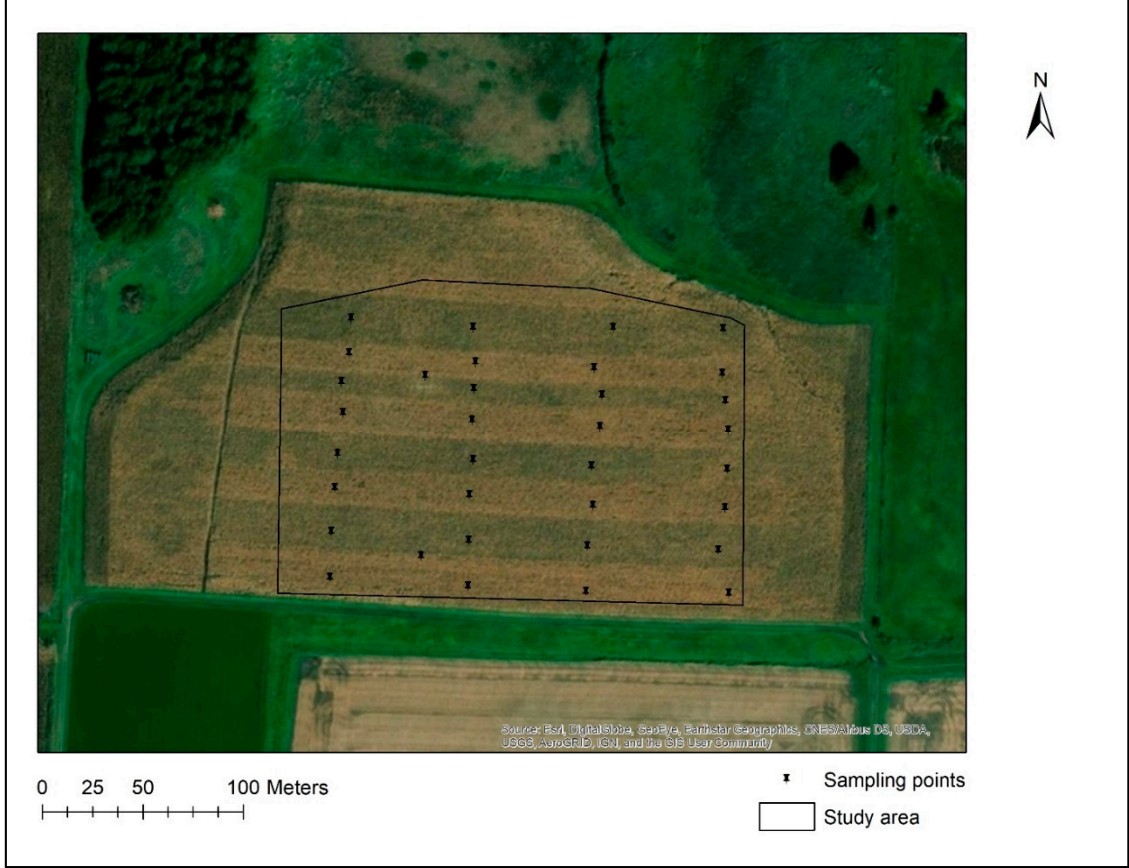

**Figure 1.** Map of the study site, which delineates the study area and shows soil and plant sampling points for the 2017 growing season.

In 2016, soil samples were collected before planting at 34 georeferenced points from the 0–30-cm, 30–60-cm, 60–90-cm, and 90–120-cm depths for analysis of soil texture. Each soil sample was composed from two sub-samples collected around each point. Soil samples from the 0–30-cm and 30–60-cm depths were analyzed for total N, ammonium-N, nitrate-N, organic carbon, cation exchange capacity, and pH. In 2017, soil samples were collected before planting from an additional 8 sampling points within the N-fertilized treatments at 0–30-cm and 30–60-cm depths for total N, ammonium-N, and nitrate-N. In both years, soil samples were air-dried and sieved to pass through a 2 mm mesh before lab analysis.

In both years, plant stand counts across 3 m lengths in three different rows were determined around each georeferenced sampling point. In 2016, five plant samples were collected within a 2-m radius of georeferenced points at the eight leaf collar (V8), tasseling (VT), and physiological maturity stages and air dried in a forced-air oven at 60 °C until reaching a constant mass for dry biomass. As in 2016, plant samples were collected in 2017 at three dates near five leaf collar (V5), ten leaf collar (V10), and VT growth stages for dry biomass. In both years, maize grain yield was measured using a field-scale combine (John Deere S600, Deere and Co., Moline, IL, USA) equipped with a yield monitoring device (John Deere StarFire$^{TM}$ 3000 GPS receiver and GreenStar$^{TM}$ 2620 display, Deere and Co., Moline, IL, USA). Yield data points for both years were processed using GreenStar$^{TM}$ Apex 3.7.9 farm management software following the John Deere S-series combine calibration guidelines (Deere and Company, Moline, IL, USA, 2012).

Daily weather data of maximum and minimum air temperature (°C), solar radiation (MJ m$^{-2}$), and rainfall (mm) were obtained from the weather station located approximately 800 m from the field.

### 2.3. Satellite Imagery and Image Processing

RapidEye Ortho level 3A satellite images were used for this study. The images are orthorectified tile products with geometric and terrain corrections in Universal Transverse Mercator map projections [39]. The RapidEye satellite used a multispectral imager with five channels sensitive between 440 and 850 nm and providing images with a spatial resolution of 5 m (Table 1).

**Table 1.** Spectral bands and their respective range of wavelengths for RapidEye imagery.

| Spectral Bands | Wavelength (nm) |
|:---:|:---:|
| Blue | 440–510 |
| Green | 520–590 |
| Red | 630–685 |
| Red Edge | 690–730 |
| Near Infrared | 760–850 |

Satellite images acquired on June 26 (Julian day 177) and July 17 (Julian day 195) in 2017 were used for this study, which corresponded to the V5 and V10 stages of maize development, respectively. Digital numbers (DN) in images were first converted to top-of-atmosphere radiance (RAD) values by multiplying the DN with the radiometric scale factor given in the metadata file of the images. The RAD values ($Wm^{-2}$ $\mu m^{-1}$ $sr^{-1}$) were then converted to reflectance (REF) values (dimensionless) using the formula [40]:

$$REF_{(i)} = RAD_{(i)} \times \frac{\pi \times SunDist^2}{EAI_{(i)} \times \cos(SolarZenith)} \qquad (1)$$

where *i* refers to corresponding band, SunDist denotes the Earth–Sun distance at the image acquisition date in astronomical units, EAI refers to exo-atmospheric irradiance and SolarZenith denotes the solar zenith angle in degrees obtained by subtracting sun elevation from 90°. SunDist values for each imagery acquisition date were obtained from Chander et al. [41]. The EAI values for each band were obtained from RapidEye [39]. The sun elevation values for each imagery acquisition date were obtained from the metadata file of the images. The REF values of spectral bands were then used to calculate vegetative indices. Previous studies have shown a strong correlation of maize growth status with NDVI and red-edge NDVI (RENDVI), [42–44]; therefore, only these two indices were considered for this study. The NDVI [22,23] and RENDVI [45,46] for each date were calculated as follows:

$$NDVI = (NIR - R)/(NIR + R) \qquad (2)$$

$$RENDVI = (NIR - RE)/(NIR + RE) \qquad (3)$$

where NIR, R, and RE denote reflectance values of near-infrared, red, and red-edge spectral bands, respectively. Regression between both indices and maize biomass measured at V5 and V10 were conducted. The best performing index based on higher coefficient of determination ($r^2$) was used for biomass estimations across the field. All image processing steps and calculations were carried out using the ModelBuilder and Raster Calculator tools in ArcGIS Desktop 10.5.1 [47].

### 2.4. CERES-Maize Model

The cropping system model (CSM) CERES-Maize [48], one of 42 crop simulation models within the Decision Support System for Agrotechnology Transfer (DSSAT) v. 4.7.5 [25,49], was used for this study. The DSSAT incorporates the dynamics of soil–plant–atmosphere interactions and simulates maize growth, development, and yield as functions of genotype, weather, soil, and crop management information [25].

### 2.4.1. Model Inputs

The minimum dataset required to run DSSAT crop models includes daily weather data, crop management information, soil profile data, and cultivar information. The minimum weather inputs are daily maximum and minimum air temperature (°C), solar radiation (MJ m$^{-2}$), and precipitation (mm). Weather data were obtained from the automated weather station located at the research site. The WeatherMan tool in DSSAT was used to prepare weather files for the CSM CERES-Maize model. Crop management information including sowing date, seeding rate, tillage, and fertilization strategy used in the study were entered through XBuild, a DSSAT tool for describing experiments. Similarly, SBuild, a tool for creating and modifying soil profile data as required for model simulations, was used to enter soil texture, total N%, ammonium-N (ppm), nitrate-N (ppm), organic carbon%, cation exchange capacity, and soil pH obtained from soil analyses. SBuild was also used to estimate the missing data for bulk density, saturated water content, field capacity, wilting point, and saturated hydraulic conductivity based on the soil texture through pedo-transfer functions. Cultivar information on days to anthesis and physiological maturity, biomass at anthesis, and yield was obtained from the fertilized treatment of the 2016 growing season.

### 2.4.2. Geospatial Data Management

The measured and estimated soil properties at each soil sampling point and depth in 2016 were associated with their respective geographic coordinates in a point shapefile. Soil properties at each sampling depth were interpolated into soil layers using ordinary kriging. Altogether, 44 soil layers were created, which included measured and estimated soil properties at different soil profile depths. Similarly, yield data points from each fertilized and unfertilized plot from the 2017 growing season were interpolated into yield layers. The interpolated yield layers were resampled to 5 m to match the spatial resolution of satellite imagery. Then, resampled yield layers were grouped into five classes using an isocluster unsupervised classification algorithm, resulting in 109 zones (51 in N-fertilized and 58 in unfertilized treatment). These zones were later used as base layer polygons for crop simulation. All kriging, resampling, and classification were carried out in ArcGIS Desktop 10.5.1 [47]. Average values of yield and each soil property for each zone were calculated using zonal statistics within the raster analysis toolbox in Quantum GIS (QGIS) 3.6.2 [50]. Average values of estimated biomass from remote sensing imagery for each zone were also obtained using the same procedure. A zonal statistics tool appended the averaged data to the respective zone in the attribute table. Soil files and crop management files required to run CSM CERES-Maize in each zone were created as required for the DSSAT file format using template files in Geospatial Simulation version 1.3 (GeoSim) [51] in Quantum GIS 3.6.2 software [50]. The simulation control tool in GeoSim was then used to run simulations in each zone, which transferred all the required geospatial data located in the attribute table from the respective zone to the model.

### 2.4.3. Model Calibration

Calibration of CSM CERES-Maize requires the estimation of six genotype-specific coefficients: P1 (thermal time from seedling emergence to end of juvenile phase), P2 (extent of delay in development for each hour with daylength above 12 h), P5 (thermal time for silking to physiological maturity), phyllochron interval between successive leaf tip appearances (PHINT), G2 (maximum possible number of kernels per plant), and G3 (kernel growth rate during grain-filling stage under optimum conditions). The first four coefficients (P1, P2, P5, and PHINT) regulate the development of maize, whereas the last two coefficients (G2 and G3) regulate yield of maize. For model calibration, phenology observations, biomass, and yield data collected from the N-fertilized treatments in 2016 were used. Generalized likelihood uncertainty estimation (GLUE), a Bayesian parameter estimation procedure within DSSAT, was used for calibration. Initially, phenology-related parameters were calibrated using anthesis and physiological maturity dates. Later, yield-specific parameters were estimated using measured

biomass and yield. At each step, GLUE was set to run for a total of 10,000 iterations to estimate the genotype coefficients.

### 2.4.4. Spatial Optimization and Data Integration

Agricultural fields are inherently heterogenous due to spatially variable soil characteristics. Yet, because of their point-based nature, crop models such as in DSSAT assume homogenous field areas. Therefore, for a field-scale crop simulation, site-specific optimization of soil parameters is required to account for spatial variability in soil properties. Currently, however, DSSAT does not have a tool for spatial optimization of soil parameters. In this study, biomass estimated from satellite imagery was used to account for the effect of the spatial variability of soil on maize growth. For this, CSM CERES-Maize was run using the simulation optimizer tool in GeoSim, which performed spatial optimization. Geosim employs a simulated annealing algorithm and facilitates the optimization process by adjusting the user-selected parameters to improve the agreement between measured and simulated data [51]. In this study, for site-specific soil parameter optimization, maize biomass estimated from remote sensing images at V5 and V10 was used (Figure 2). Soil parameters chosen for optimization were total soil N concentration (SLNI) in the top 30 cm of soil and soil fertility factor (SPLF). As the research involved N-fertilized and unfertilized treatments and as N is a highly mobile nutrient, spatial processes such as leaching and runoff could have changed the soil N level. To account for spatial variability in soil N level, SLNI was selected. The SLPF was chosen to address soil fertility issues other than N that were not accounted for in the model. Based on soil analyses, initial values of SLNI were set at 0.2% and 0.1% for N-fertilized and unfertilized treatments, respectively. The SLPF was set to 1 for both treatments. For optimization, SLNI was allowed to vary from 0.01% to 0.4%, whereas SLPF was allowed to vary from 0.7 to 1. Both parameters were optimized independently as well as simultaneously to reduce the error between estimated biomass from satellite imagery and biomass simulated by the model. The optimization involved initialization of SLNI and SLPF with the optimized values.

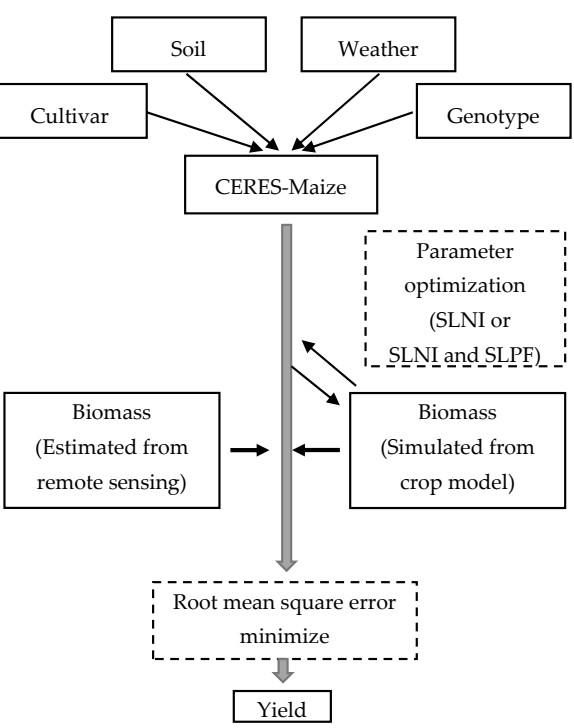

**Figure 2.** Flow chart of using biomass data estimated from satellite imagery into the CERES-Maize model for spatial optimization of total soil nitrogen concentration (SLNI) or SLNI and soil fertility factor (SLPF) (boxes in solid lines indicate inputs and outputs and boxes in dashed lines indicate optimization process).

### 2.4.5. Model Evaluation

The model was evaluated for phenology, biomass, and grain yield data collected during the 2017 growing season. Two separate evaluations were conducted, without and with site-specific optimization of soil properties. For evaluation, root mean square error (RMSE) and normalized RMSE (nRMSE) were used. The RMSE between simulated and measured values was calculated as follows:

$$RMSE = \sqrt{\frac{\sum_{i=1}^{n}(S_i - M_i)^2}{n}} \qquad (4)$$

where *n* refers to number of observations, *S* and *M* denote simulated and measured values, respectively. Then, nRMSE was calculated as RMSE / (mean of measured values). Lower RMSE and nRMSE values indicate better model performance.

## 3. Results and Discussion

### 3.1. Weather Conditions during the Growing Seasons

Growing seasons during both study years received more rainfall than the historical average (Table 2). Total cumulative rainfall was 723 and 651 mm during the 2016 and 2017 growing seasons, respectively. In both years, May was wetter, whereas June was drier than the historical average (Table 2).

**Table 2.** Comparison of 2016 and 2017 monthly average air temperature (Tavg; °C) and total rainfall (Rain; mm) with historical averages at Lamberton, Minnesota, the United States.

| Month | Historical Average (1961–2014) | | Deviation from Historical Average [‡] | | | |
|---|---|---|---|---|---|---|
| | | | 2016 | | 2017 | |
| | **Tavg** | **Rain** | **Tavg** | **Rain** | **Tavg** | **Rain** |
| May | 14.4 | 88 | +0.3 | +53 | −0.8 | +64 |
| June | 20.2 | 104 | +1.2 | −38 | +0.5 | −36 |
| July | 22.5 | 91 | −0.4 | +85 | −0.1 | +11 |
| Aug | 20.5 | 80 | +0.9 | +55 | −1.7 | +45 |
| September | 16.1 | 78 | +1.6 | +55 | +1.6 | -24 |
| October | 9.2 | 52 | +0.9 | +20 | +0.3 | +98 |

[‡] "+" denotes above and "−" denotes below historical average.

Considering the historical average of the growing season, 2016 was warmer by 0.75 °C, whereas 2017 was cooler by 0.02 °C. During both years, June, September, and October were warmer, whereas July was cooler (Table 2). August in 2017 was also cooler by 1.7 °C than the historical average. Monthly average air temperature during the 2016 growing season was often higher than in 2017 (Figure 3).

### 3.2. Relation between Vegetative Indices and Maize Biomass

At the V5 stage, measured maize biomass ranged from 112 to 519 kg ha$^{-1}$, whereas at the V10 stage, maize biomass ranged from 1343 to 5079 kg ha$^{-1}$. The $r^2$ between biomass and RENDVI at the V5 and V10 stages was 0.44 and 0.58, respectively, whereas $r^2$ with NDVI was 0.55 and 0.63, respectively (Figure 4).

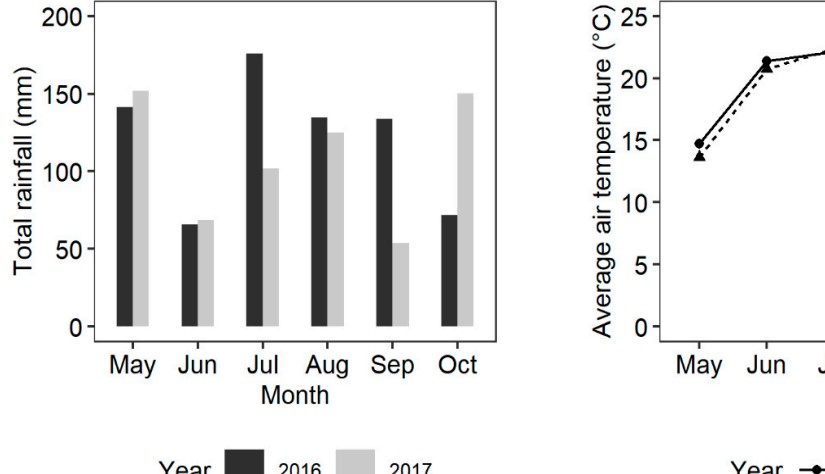

**Figure 3.** Total monthly rainfall and monthly average air temperature during 2016 and 2017 growing seasons at the study site.

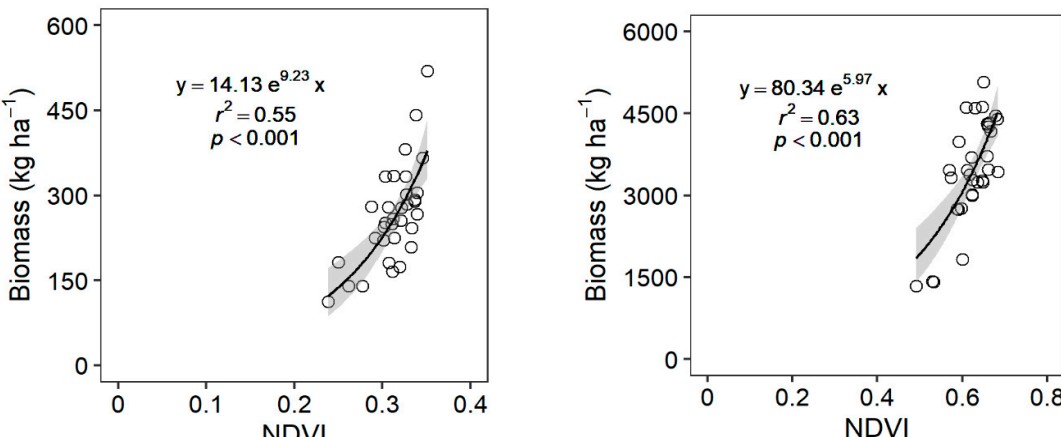

**Figure 4.** Relationship between normalized difference vegetation index (NDVI) and maize biomass at five (**left**) and ten (**right**) leaf collar growth stages.

At both stages, NDVI had greater $r^2$ than RENDVI. For both indices, greater $r^2$ was achieved at V10 than at V5. The NDVI values increased with the growth stages; from 0.24 to 0.35 at V5 and from 0.49 to 0.68 at V10. At both stages, lower and higher NDVI readings were associated with lower and higher biomass, respectively (Figure 4).

The predictive ability of NDVI to explain maize biomass variability has been well documented [42–44]. As observed in this study, increase in NDVI as maize grows and develops has also been reported by Shaver et al. [52] and Martin et al. [43]. However, as maize growth progresses, NDVI tends to saturate after canopy closure. In this situation, NDVI can have lower explanatory ability to distinguish biomass variability [53]. As the red edge region of the electromagnetic spectrum can penetrate deeper into the crop canopy due to lower chlorophyll absorption, red-edge-based indices such as RENDVI have been shown to better explain growth variability at later growth stages [53,54].

The regression equations for the relationship between NDVI and biomass (Figure 4) were used to estimate maize biomass for all pixels in the RapidEye imagery. The spatial variability of estimated biomass ranged from 105 to 714 kg ha$^{-1}$ at V5 (Figure 5) and from 1209 to 5775 kg ha$^{-1}$ at V10 (Figure 6). The ranges of estimated biomass did not vary greatly with the measured biomass. The use of vegetative indices calculated from remote sensing data to estimate crop growth parameters is frequent in literature. For example, Kross et al. [44] used RapidEye imagery to calculate several vegetative indices, including NDVI and RENDVI, to estimate LAI and biomass of maize and soybean (*Glycine max* L. Merr.). Parallel

to the finding of this study, Kross et al. [44] also found NDVI better in estimating biomass as compared to other indices. Gitelson et al. [55] proposed new indices based on near infrared, red, and green bands of the spectrum for improved estimation of LAI and leaf biomass in maize. In addition to these vegetation indices, physical models involving physical laws and the inversion of canopy reflectance spectra are also used [33]. For example, Thorp et al. [56] used the inversion of a radiative transfer model to estimate wheat growth parameters that affected LAI and biomass in the CERES-Wheat model. Unlike statistical models, physical models require estimation of several variables of soil, leaf, and canopy to simulate the reflectance at leaf and canopy levels [33].

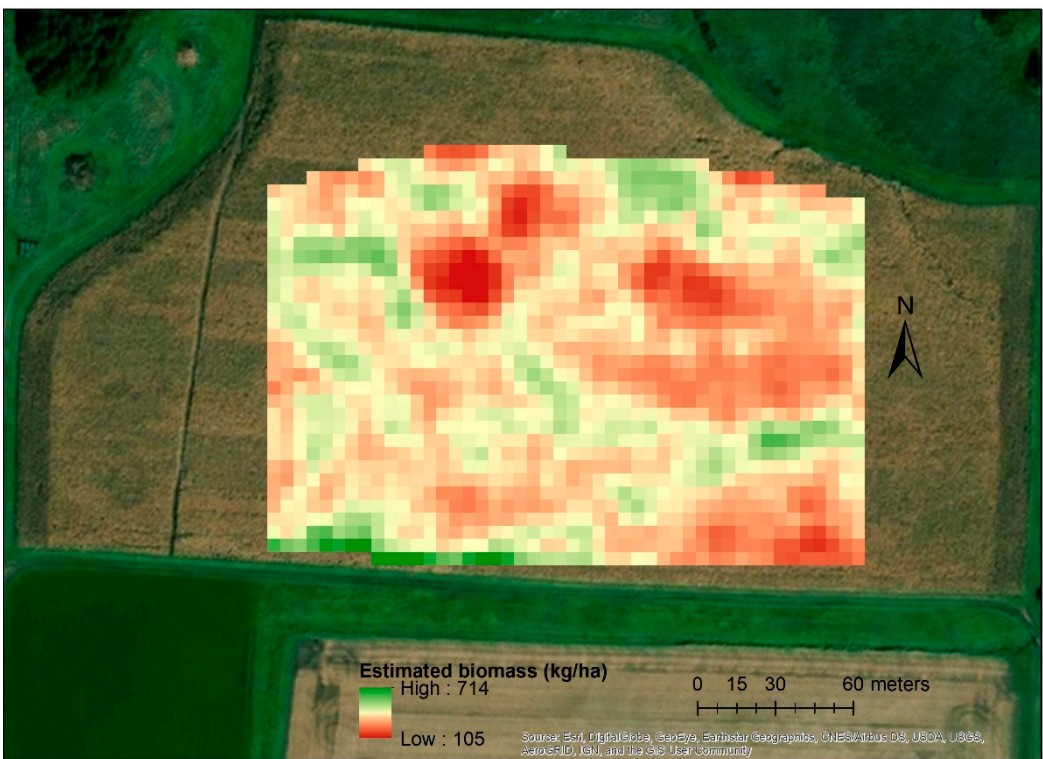

**Figure 5.** Spatial variability in maize biomass at the five leaf collar stage estimated using NDVI values from RapidEye satellite imagery.

*3.3. Model Calibration Genetic Coefficients*

The calibrated coefficients obtained (Table 3) gave satisfactory simulation of phenology. Days to anthesis was accurately predicted within one day of the observed results and days to physiological maturity was also accurately predicted. During calibration, biomass was simulated with a RMSE of 2312 kg ha$^{-1}$ at anthesis and RMSE of 1678 at physiological maturity; yield was simulated with a RMSE of 2175 kg ha$^{-1}$.

*3.4. Model Evaluation with and without Spatial Optimization*

Upon evaluation with 2017 data, the calibrated model simulated anthesis and physiological maturity accurately. Physiological maturity in 2017 was delayed and occurred in late September (as compared to the usual time in early September) due to cooler than average air temperatures in July and August (Table 2), and the model correctly captured this delay in physiological maturity.

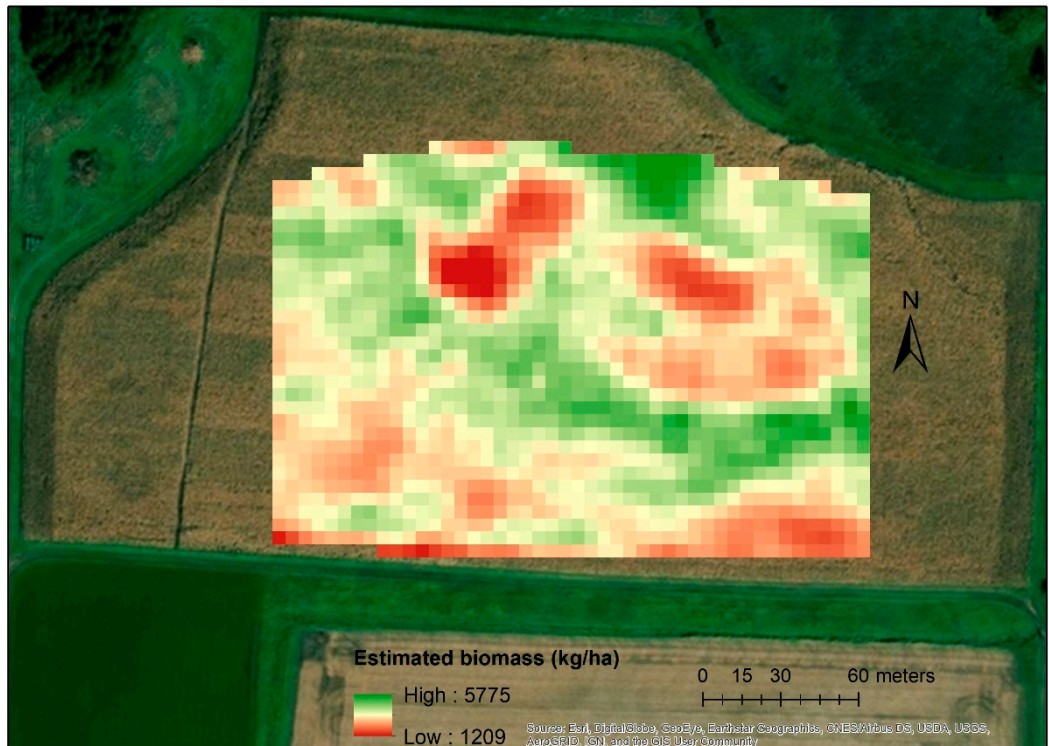

**Figure 6.** Spatial variability in maize biomass at ten leaf collar stage estimated using NDVI values from RapidEye satellite image.

**Table 3.** Calibrated values of cultivar coefficients for CERES-Maize obtained using the generalized likelihood uncertainty estimation procedure in Decision Support System for Agrotechnology Transfer (DSSAT) v. 4.7.5.

| Maize Cultivar Coefficient | Calibrated Values |
|---|---|
| Thermal time from seedling emergence to the end of the juvenile phase (P1) | 159.7 |
| Extent to which development is delayed for each hour that daylength is above 12.5 h (P2) | 1.409 |
| Thermal time for silking to physiological maturity (P5) | 669.4 |
| Phyllochron interval between successive leaf tip appearances (PHINT) | 38.9 |
| Maximum possible number of kernels per plant (G2) | 875.3 |
| Kernel growth rate during linear grain filling stage under optimum conditions (G3) | 8.7 |

Without spatial optimization, the model overpredicted maize biomass at V5 with RMSE of 264 kg ha$^{-1}$ (nRMSE 101%), which was more than twice the average measured biomass. The results were similar for the N-fertilized and unfertilized treatments (Table 4). Site-specific optimization of SLNI based on V5-estimated biomass slightly reduced the simulated biomass RMSE to 245 kg ha$^{-1}$ (nRMSE 94%). Simultaneous adjustment of SLNI and SLPF did not further improve the simulation (Table 4). Improvement in biomass simulations after SLNI optimization was greater in the N-fertilized treatment, which reduced the RMSE from 275 to 236 kg ha$^{-1}$ (nRMSE from 102% to 88%).

**Table 4.** Root mean square error (RMSE; kg ha$^{-1}$) and normalized RMSE (nRMSE; %) between measured and simulated maize biomass in nitrogen (N)-fertilized and unfertilized treatments at five (V5) and ten leaf collar (V10) stages, with and without spatial optimization.

| Optimization [‡] | Average | N-Fertilized | N Unfertilized |
|:---:|:---:|:---:|:---:|
| | | RMSE (nRMSE) | |
| | | Biomass at V5 | |
| None | 264 (101) | 275 (102) | 253 (99) |
| SLNI | 245 (94) | 236 (88) | 253 (99) |
| SLNI and SLPF | 245 (94) | 236 (88) | 252 (99) |
| | | Biomass at V10 | |
| None | 1255 (36) | 654 (19) | 1608 (46) |
| SLNI | 1094 (31) | 617 (18) | 1383 (39) |
| SLNI and SLPF | 969 (28) | 507 (14) | 1240 (36) |

[‡] None refers to without any spatial optimization. SLNI refers to optimization of total soil nitrogen concentration (SLNI) only. SLNI and SLPF refer to simultaneous optimization of SLNI and soil fertility factor (SLPF).

Poor simulations of maize biomass at V5 could have been improved by better calibration with more in-season biomass samplings. Also, maize biomass data from multiple sites or years could have given more accurate cultivar coefficients during calibration to simulate biomass at early growth stages. The only slight improvement in simulation after optimization of SLNI and SLPF is also due to the calibration of genotypic-specific coefficients or any crop parameters rather than soil parameters. Another possible reason could have been due to earlier emergence of maize in the model than in the actual field conditions, which led to greater biomass simulations at V5. Previous studies have used remote sensing imagery to calibrate genotype specific coefficients and crop management parameters. Thorp et al. [17], for example, used LAI estimates from NDVI readings to spatially optimize specific leaf area of cultivar (SLAVR) of the DSSAT–CROPGRO-Cotton model, which regulates potential specific leaf area in cotton (*Gossypium hirsutum* L.). Similarly, Li et al. [36] used remotely estimated LAI and canopy nitrogen content in a DSSAT–CERES-Wheat model to optimize several crop genotype-specific parameters, including PHINT, and crop management parameters, such as plant population. However, our study did not consider optimization of any genotype-specific parameters and only optimized soil-related parameters to account for spatial variability in soil properties.

Considering nRMSE results, the biomass simulation results were comparatively better at the V10 stage than at V5. Without spatial optimization, biomass at the V10 stage was simulated with RMSE of 1255 kg ha$^{-1}$ (nRMSE 36%) (Table 4). Biomass in the N-fertilized treatment (nRMSE 19%) was simulated better than in the unfertilized treatment (nRMSE 46%). The RMSE of N-fertilized simulated biomass was 654 kg ha$^{-1}$, whereas the unfertilized treatment was 1608 kg ha$^{-1}$. Biomass in the unfertilized treatment was underpredicted because of a greater N stress simulation by the model.

Following site-specific optimization of SLNI using V10-estimated biomass, the overall RMSE for simulated biomass was reduced by 12.8%, from 1255 to 1094 kg ha$^{-1}$. Simultaneous optimization of SLNI and SLPF reduced the RMSE by 22.7% to 969 kg ha$^{-1}$. The optimization improved the biomass simulation in both N-fertilized and unfertilized treatments. Optimization of only SLNI reduced the RMSE from 654 to 617 kg ha$^{-1}$ in the fertilized treatment. The RMSE was reduced even lower to 507 kg ha$^{-1}$ when both SLNI and SLPF were optimized simultaneously (Table 4). The trend was similar in the unfertilized treatment. Optimizing SLNI alone reduced the RMSE from 1608 to 1383 kg ha$^{-1}$ in the unfertilized treatment. Simultaneous optimization of both SLNI and SLPF further reduced the RMSE to 1240 kg ha$^{-1}$. Thus, optimization of SLNI resulted in 5.6% and 13.9% reduction in RMSE in the N-fertilized and unfertilized treatments, respectively, whereas simultaneous optimization of both SLNI and SLPF resulted in 22.8% and 22.4% reduction in RMSE in N-fertilized and unfertilized treatments, respectively. Following optimization, greater reduction in RMSE in unfertilized treatments

could be from the improvement in the representation of soil N level, which eventually decreased the N stress simulation in the model.

The average measured maize grain yield ranged from 10,547 to 11,009 kg ha$^{-1}$ among N-fertilized strips and 4393 to 5984 kg ha$^{-1}$ among unfertilized strips (Table 5). Without spatial optimization, overall maize yield was simulated with RMSE of 1264 kg ha$^{-1}$ (nRMSE 15.7%) (Figure 7). As with biomass, yield in the N-fertilized treatment was also simulated better than the unfertilized treatment. The RMSE of simulated yield in the N-fertilized treatment was 1132 kg ha$^{-1}$ (nRMSE 10%), whereas in the unfertilized treatment it was 1370 kg ha$^{-1}$ (nRMSE 25%). The coefficients of variation of the average simulated yield without optimization were less than that of the measured yield in both N-fertilized and unfertilized strips (Table 5), indicating that soil properties were more homogenous in the model than in the actual field conditions.

**Table 5.** Average measured and simulated maize yield (kg ha$^{-1}$) with coefficient of variation (%; in parenthesis) of nitrogen (N)-fertilized and unfertilized strips, with and without spatial optimization.

| Yield [‡] | F1 [¥] | F2 | F3 | F4 | UF1 [¥] | UF2 | UF3 | UF4 |
|---|---|---|---|---|---|---|---|---|
| Measured | 10,861 (5.0) | 11,009 (3.1) | 10,858 (3.4) | 10,547 (5.6) | 5984 (7.6) | 5903 (20.7) | 4393 (30.0) | 5719 (15.2) |
| No optimization | 11,966 (2.3) | 11,945 (1.5) | 11,864 (1.3) | 11,799 (2.2) | 4652 (3.6) | 4705 (2.3) | 4627 (3.9) | 4735 (2.0) |
| V5 SLNI | 11,568 (1.1) | 11,577 (1.0) | 11,530 (1.0) | 11,534 (1.3) | 4652 (3.5) | 4705 (2.4) | 4627 (3.9) | 4735 (2.0) |
| V5 SLNI and SLPF | 10,452 (1.2) | 10,681 (1.1) | 11,350 (5.5) | 11,060 (8.0) | 4626 (3.7) | 4678 (2.4) | 4625 (3.9) | 4686 (2.1) |
| V10 SLNI | 11,603 (1.8) | 11,837 (2.3) | 11,702 (1.9) | 11,727 (2.4) | 5091 (4.4) | 5119 (3.7) | 4860 (8.4) | 5102 (4.9) |
| V10 SLNI and SLPF | 9750 (10.8) | 11,378 (9.3) | 10,852 (11.1) | 10,741 (12.6) | 5069 (4.4) | 5076 (4.5) | 4819 (9.0) | 5074 (5.7) |

[‡] No optimization refers to without any spatial optimization. V5 and V10 refers to optimization performed using estimated biomass at five and ten leaf collar stages, respectively. SLNI refers to optimization of total soil nitrogen concentration (SLNI) only. SLNI and SLPF refers to simultaneous optimization of both SLNI and soil fertility factor (SLPF); [¥] F1 to F4 represent N-fertilized strips, whereas UF1 to UF4 represent N-unfertilized strips.

Spatial optimization of SLNI from data integration at the V5 and V10 stages improved the overall yield simulations (Figure 7). However, slightly better yield simulations were obtained from optimization of SLNI at V10 (nRMSE 12.8%) as compared to V5 (nRMSE 14.4%). Only optimizing SLNI using V5-estimated biomass reduced the overall RMSE by 8.4% from 1264 to 1158 kg ha$^{-1}$. Simultaneous optimization of SLNI and SLPF slightly increased the RMSE from 1264 to 1271 kg ha$^{-1}$. Following optimization of SLNI at V5, the improvement in yield simulations were from fertilized strips, which lowered the over-predicted yield. Yield simulations in unfertilized strips, however, remained unchanged (Table 5). In addition to decreasing the yield in all strips, simultaneous optimization of SLNI and SLPF at V5 increased the coefficients of variation of average simulated yields. This increment in coefficients of variation indicates the creation of heterogeneous soil conditions in the model in terms of soil N concentration and soil fertility level.

Spatial optimization of SLNI only using V10-estimated biomass reduced the RMSE by 18.8% and gave the lowest RMSE of 1026 kg ha$^{-1}$. Simultaneous optimization of SLNI and SLPF using V10- estimated biomass produced a RMSE of 1058 kg ha$^{-1}$, which was slightly more than from the optimization of SLNI alone. With biomass at both V5 and V10, simultaneous optimization of SLNI with SLPF did not give better yield simulation as compared to single optimization of SLNI. After spatial optimization of SLNI, improvement in yield simulation occurred in both the N-fertilized and unfertilized treatments. Improvement in the N-fertilized treatment occurred from optimization at both V5 and V10, whereas improvement in the unfertilized treatment occurred only from optimization at the V10 stage. In addition to improvement in yield simulations, optimization of SLNI at V10 improved the coefficients of variation of the average simulated yield of all unfertilized strips. This improvement

in coefficients of variation showed the enhancement in representing spatial heterogeneity in soil N concentration in the model.

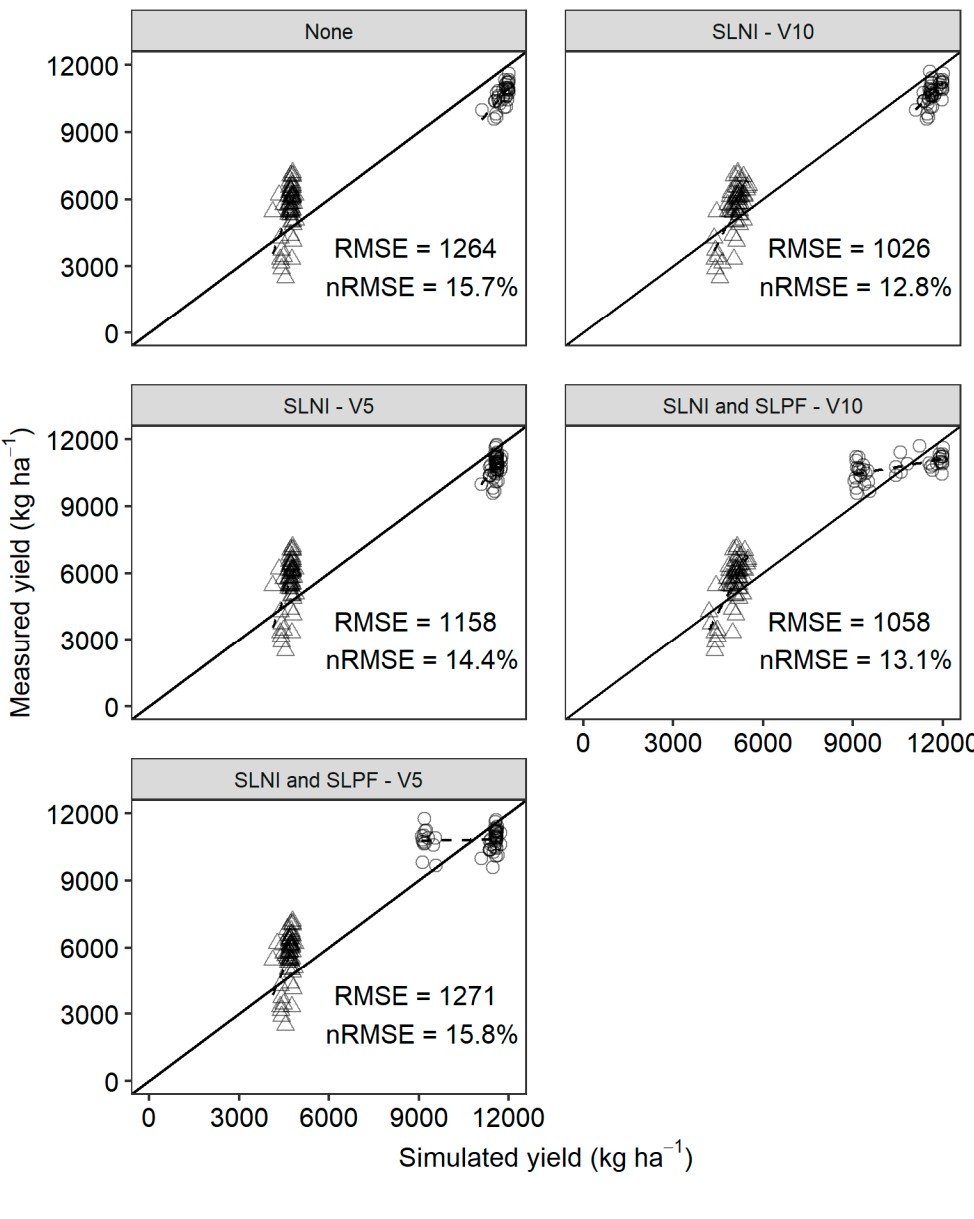

**Figure 7.** Scatterplots of simulated and measured maize yield. The diagonal black line is the 1:1 line. The dashed black line represents the linear regression between simulated and measured yields.

After data integration, the overall improvement in yield simulation was due to better representation of soil N levels in the field. Estimation of biomass at the V10 stage from satellite imagery depicted the spatial variability in maize growth. As the study site has an elevation ranging from 337 to 345 m with a downward slope towards the east (Figure 8), it is likely that zones at lower areas may have received run-off water together with N and higher levels of organic matter from upland N-fertilized treatment areas. As reported by Batchelor et al. [16] and Fraisse et al. [57], enhancement in crop models for accurate site-specific yield estimation requires consideration of spatial events, such as runoff from uplands. Although this study did not calculate any two- or three- dimensional movement of water or nutrients between the spatial zones, this study has shown an option for point-based simulation to account for the effects of these spatial events at the field scale. Accounting for such

spatial events in the crop model not only improves the estimation of site-specific yield, but also enhances the decision-making process to address spatial and temporal variability of the soil–crop relationship within the field. Understanding of this variability within the field is essential for making make informed decisions on crop management for greater economic efficiency and for addressing environmental problems. As the variability in the soil–crop relationship within the field eventually results in differences in crop yield, early estimation of spatially-explicit crop yield at the field scale can aid in precision application of agricultural inputs.

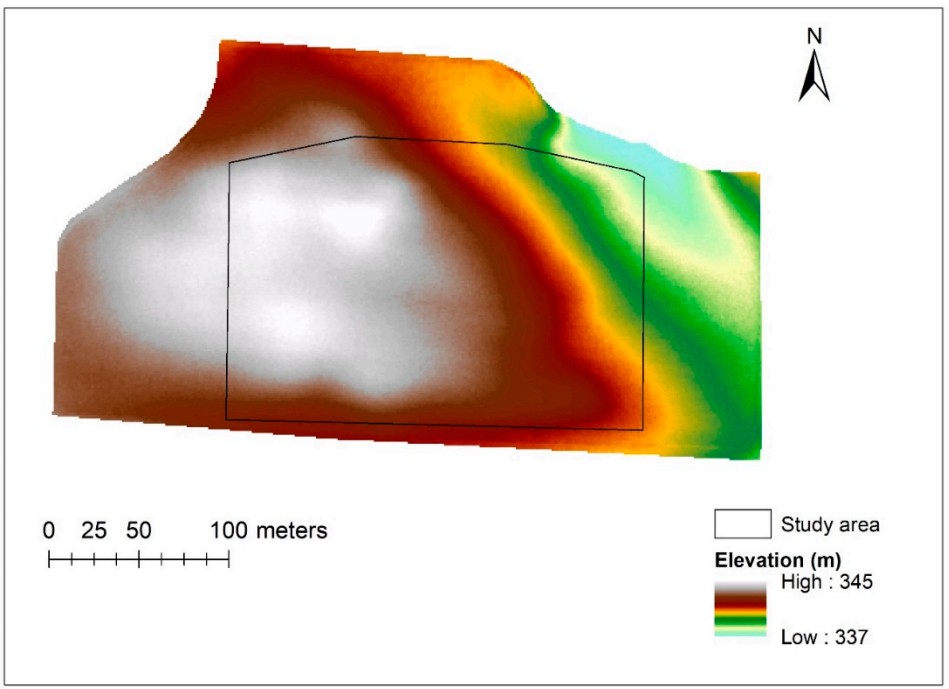

**Figure 8.** Digital elevation map of the study site. (Source: Minnesota Department of Natural Resources and Minnesota Geospatial Information Office, (MnTOPO) [58]).

## 4. Conclusions

This study evaluated the accuracy of the site-specific maize yield estimation of the stand-alone CERES-Maize model with a data integration approach, in which satellite multispectral images were used to optimize soil parameters in CERES-Maize. Spatial optimization of SLNI using estimated biomass at both V5 and V10 stages improved the overall yield simulations. More accurate yield estimations, however, were obtained from optimization at V10 as compared to V5 stage. Subsequent optimization of SLNI and SLPF did not further improve the yield simulations. This study shows that integrating remote sensing data into a crop model better depicts the whole soil–crop relationship within a field as compared to the crop model alone. This integrated approach is promising for site-specific maize yield estimation and for planning spatially variable management of agronomic inputs such as fertilizers and irrigation, aiming at improving resource use efficiency in crop production. Future studies on the use of higher-resolution images for site-specific optimization would further enhance the application of crop models in precision agriculture. Further research on crop model uncertainties due to soil inputs at the field scale and from different spatial interpolation methods used to estimate soil properties would help in improving soil data collection and crop model applications at large scale.

**Author Contributions:** V.R.J., J.A.C., and A.G.y.G. conceptualized and designed the research. V.R.J. conducted the field experiments and collected the data. K.R.T., J.A.C., and A.G.y.G. supervised the methodology and data analyses. A.G.y.G., G.A.J., P.M.P., and J.S.S. supervised the study and field experiments. V.R.J. wrote the draft of the manuscript. K.R.T., J.A.C., G.A.J., P.M.P., J.S.S., and A.G.y.G. edited and reviewed the manuscript.

**Funding:** This research received no external funding.

**Acknowledgments:** We are grateful to Farmers Edge^TM for providing in-season RapidEye satellite imagery. We appreciate the help from Lindsey Englar, Nathan Dalman, summer interns, and staff at the University of Minnesota Southwest Research and Outreach Center. We also appreciate the help from Emily Ann Evans, Communications Coordinator of the University of Minnesota Research and Outreach Centers, for proofreading the manuscript.

**Conflicts of Interest:** The authors declare no conflict of interest.

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
