# Peer review of "Improving Site-Specific Maize Yield Estimation by Integrating Satellite Multispectral Data into a Crop Model"

_agronomy, doi:10.3390/agronomy9110719_

Round 1
Reviewer 1 Report
The manuscript # agronomy-607324, "Improving site-specific maize yield estimation by integrating satellite multispectral data into a crop model", describes an approach to make the best use of remote sensing data for improvement of crop models at a site-specific scale. The method that authors have proposed would improve the in-season forecast of crop yield in a region where remote-sensing images become available readily. Still, some of the references are relatively old. Thus, it is recommended to add the recent ones such as Ban et al. (2017, remote sensing, https://doi.org/10.3390/rs9010016.)
Reviewer 2 Report
The paper could have been written with nearly identical conclusions more than 20 years ago. The CERES results are so far removed from observed yields that I must question whether the calibration was adequate. The quality of the English is poor and in need of heavy editing. I regret that I am unable to recommend that this paper be published in its current form.
Round 2
Reviewer 2 Report
Thank you for your responses.